# A Novel Pulsed Eddy Current Criterion for Non-Ferromagnetic Metal Thickness Quantifications under Large Liftoff

**DOI:** 10.3390/s22020614

**Published:** 2022-01-13

**Authors:** Haowen Wang, Jiangbo Huang, Longhuan Liu, Shanqiang Qin, Zhihong Fu

**Affiliations:** 1School of Robot Engineering, Yangtze Normal University, Chongqing 408100, China; 19990002@yznu.edu.cn (J.H.); 20190025@yznu.edu.cn (S.Q.); 2School of Electrical Engineering, Chongqing University, Chongqing 400044, China; 20133954@cqu.edu.cn (L.L.); fuzhihong@cqu.edu.cn (Z.F.)

**Keywords:** pulsed eddy current, thickness quantifications, dynamic apparent time constant, lift-off variations

## Abstract

The pulsed eddy current (PEC) inspection is considered a versatile non-destructive evaluation technique, and it is widely used in metal thickness quantifications for structural health monitoring and target recognition. However, for non-ferromagnetic conductors covered with non-uniform thick insulating layers, there are still deficiencies in the current schemes. The main purpose of this study is to find an effective feature, to measure wall thinning under the large lift-off variations, and further expand application of the PEC technology. Therefore, a novel method named the dynamic apparent time constant (D-ATC) is proposed based on the coil-coupling model. It associates the dynamic behavior of the induced eddy current with the geometric dimensions of the non-ferromagnetic metallic component by the time and amplitude features of the D-ATC curve. Numeral calculations and experiments show that the time signature is immune to large lift-off variations.

## 1. Introduction

The pulsed eddy current (PEC) inspection is a common method that is used for the detection of metal thickness. As a non-contact detection method, the PEC inspection uses a square wave with a certain duty cycle as the excitation current, which provides a wider investigation depth than conventional eddy current testing [1,2]. In recent years, PEC inspection has been proved to be an effective method for thickness detection and defect diagnosis [3,4,5,6].

The PEC inspection injects a pulse current into the transmitter coil to excite the pulsed magnetic field, which generates transient eddy current in the test pieces, and the corresponding response signal is usually collected by the receiver coil. In general, the PEC inspection investigates geometric dimensions of the metallic components based on the time-domain features of the signal. For example, it could be the peak amplitude (PA), the time to peak feature, the moment the signal is crossing by a certain level, the time interval between certain nodal points, the maximum values of amplitude, etc. [7,8,9].

However, these signal features for non-ferromagnetic metal are on the condition of a small lift-off (usually a few millimeters). Compared with the small lift-off condition, both the original and differential signal under the large lift-off condition show distinct characteristics, so signal features such as signal slopes or PA are not suitable for thickness detection of samples coated with thick insulating layers, which can be commonly found in thermal and liquefied natural gas pipelines [10].

On the other hand, conventional time-domain features of PEC are prone to suffer from lift-off effect (variations in the distance between the sensor and the target), which is an important factor limiting eddy current signals interpretation [11]. Numerous methods have been proposed to lessen the lift-off effect for effective PEC. Ref. [12] performed normalization and difference operation on sensor signals to mitigate lift-off effect for PEC evaluation, but it is not suitable for thickness measurement yet. Time to the peak of difference PEC signals is a candidate to reduce the lift-off effect [13]. However, time-to-peak signature is likely subjected to noises in PEC signals, which poses an adverse impact on measurements. Meanwhile, the criterion based on the signal decay rate has low dependence on lift-off [6,14], but the thickness feature extracted from the later stage of the signal is inevitably interfered with by the system noise.

As a special signal feature, the lift-off point of intersection (LOI) that was originally proposed by D.L. Waidelich, is a point where PEC signals intersect when only liftoff distance varies. Basically, LOI points are immune to lift-off variations but dependent on the properties of the objects under inspection. Ref. [15] investigated and interpreted the behaviors of LOI signature due to a plate with varying conductivity and thickness in physics and found the nonlinear relationship between the sample thickness and the amplitude and time of LOI points. In fact, the sensitivity of LOI signature decreases rapidly with the increase of specimen thickness. Therefore, its application is limited to the thickness detection of non-ferromagnetic coatings [16]. Not only that, current LOI investigations are still on the condition of millimeter-level lift-off.

Ref. [10] proposed the time to the last peak point (TLPP) to realize non-ferromagnetic metal thickness detection under large lift-off conditions. However, the TLPP of the thin metal plate is susceptible to the interference of the system noise. This is because the reduction of the sample thickness delays the time to TLPP while reducing the amplitude of the TLPP. Meanwhile, the author pointed out that wall thinning calculation based on a certain lift-off is inaccurate, and errors will be inevitably produced due to the changeable lift-off. Therefore, new signal features should be proposed to adapt to the non-ferromagnetic metallic component under the large lift-off variations.

This paper proposes a new signal feature named the dynamic apparent time constant (D-ATC) based on the full-stage of the PEC response in the coil-coupling model. It associates the dynamic behavior of the induced eddy current with the geometric dimensions of the test pieces by the time and amplitude signatures of the D-ATC curve. As a normalized decay rate indicator, the D-ATC of eddy current is insensitive to the lift-off. Therefore, it can be used to detect the thickness of aluminum plate under large lift-off variations.

The breakdown of this paper is as follows: the next section introduces the concept of dynamic time constant with the coil coupling model and provides an algorithm for calculating the D-ATC curve from the conventional PEC response. The correlation between the D-ATC curve and the geometric dimensions of the specimen is later analyzed in Section 3 by employing a transient electromagnetic model. The characterization of the aluminum plate thickness in the D-ATC curve is described in Section 4. The impact of lift-off variations on the conventional and the D-ATC method is discussed in Section 5. The above simulation results are supported by experimental data shown in Section 6.

## 2. Apparent Time Constant of the PEC Response

During the interval of pulse excitation, the eddy current in the test piece diffuses similar to a smoke ring while decays under the action of Joule dissipation, thus the general eddy current effect of the specimen can be approximated as an equivalent current loop with finite dimensions, as shown in Figure 1a. 

The dynamic behavior of this equivalent current loop can be analyzed through a coil-coupling model (CCM), which was first put forward by Loos in 1976 [17] and revised by Tan et al. and Chen et al. [18,19]. 

The CCM applies an air-core transformer model to approximate the traditional eddy current testing model, as shown in Figure 1b, in which the primary side is the excitation circuit, and the secondary side consists of an inductor *L* and a resistor *R* derived from the equivalent current loop.

Therefore, the dynamic behavior of the equivalent current loop follows the time constant τ=L/R of the inductive first-order circuits, in which *L* can be calculated by
(1)$$L=\mu_{0}N^{2}r_{e}\left\{\left[1+\frac{{\left(\frac{r_{c}}{r_{e}}\right)}^{2}}{8}\right]\log\left(\frac{8r_{e}}{r_{c}}\right)-1.75+\frac{{\left(\frac{r_{c}}{r_{e}}\right)}^{2}}{24}\right\}$$ in which $\mu_{0}$ is the permeability of vacuum, $r_{c}$ and N represent the section radius and equivalent turns of ECD respectively, and $r_{e}$ is projection radius of the maximum of ECD on the ground. 

The resistance R of the equivalent current loop can be expressed as
(2)$$R=\frac{2r_{e}}{\sigma {r_{c}}^{2}},\;$$ in which σ is the conductivity of the medium.

To calculate the equivalent current loop’s dimensions, the equivalent current density is selected to be equal to the maximum eddy current density on the specimen surface. When the excitation current is completely turned off (t=0), the maximum value of eddy current appears near the edge of the outer diameter ($r_{2}$) of the excitation coil [20]. At time t, the projection radius $r_{e}$ of the maximum of eddy current density (ECD) on the ground is
(3)$$r_{e}=r_{2}+\sqrt{C_{1}t\sigma /\mu },\;$$ and the penetration depth *D* in uniform half space is deduced as [21]
(4)$$D=\sqrt{\frac{t}{C_{2}\sigma \mu }},$$ in μ is the permeability of the medium, respectively, and $C_{1}$ and $C_{1}$ is a constant.

Considering the diffusion behavior of ECD that shown by (1) and (2), $r_{e}$ and $r_{c}$ are variables about time, and so is the corresponding time constant. By employing the revised CCM [21], the PEC induced detector coil voltage $\varepsilon_{1}\left(t\right)$ can be modelled as an infinite summation of exponential terms, such as
(5)$$\varepsilon_{1}\left(t\right)=\sum_{n=1}^{\infty }A_{n}\exp\left(-\frac{t}{\tau_{n}}\right),$$ in which $A_{n},\tau_{n}\in R,\;\tau_{n}>0$ for all  n. 

For an exponential decay signal $\varepsilon \left(t\right)=Ae^{-t/\tau }$, its time constant τ can be extracted by the differentiation operations shown as
(6)$$\tau =\frac{\varepsilon \left(t\right)}{-d\varepsilon \left(t\right)}=\frac{Ae^{-t/\tau }}{\frac{A}{\tau }e^{-t/\tau }},$$ or by the integral operations (7)$$\tau =\frac{-\int \varepsilon \left(t\right)dt}{\varepsilon \left(t\right)}=\frac{\tau Ae^{-t/\tau }}{Ae^{-t/\tau }},\;$$ which has better adaptability to system noise.

Taking into account the dispersion and diffuseness of induced eddy currents in the inspected material, the characterization used to evaluate the attenuation rate of the PEC response should be called the apparent time constant $\tau_{ATC}$,

Therefore, when (7) is applied to the PEC induced detector coil voltage $\varepsilon_{1}\left(t\right)$, the apparent time constant $\tau_{ATC}$ is a variable of time, as shown by
(8)$$\tau_{ATC}\left(t\right)=\frac{-\int \varepsilon_{1}\left(t\right)dt}{\varepsilon_{1}\left(t\right)}$$

Considering that the actual sampling signal is formed by discrete data, the numerical integration range is usually set to cover several data near the integration point.

In summary, the apparent time constant of the PEC response is a time-dependent variable. This dynamic apparent time constant (D-ATC) curve reflects the dynamic behavior of the induced eddy current in the inspected material.

## 3. The D-ATC Method

This section takes a transient electromagnetic model to show the general shape of the D-ATC curve and analyze its correlation with the dynamic behavior of ECD.

As shown in Figure 1a, The PEC inspection model is implemented with a metal plate and a center loop device. As the inspection coil probe, the central loop device located on the z=100 mm plane consists of two separate coils formed by a transmitter and a coaxial receiver coil, their radii are 50 mm and 20 mm, respectively. An aluminum plate of 300 mm × 200 mm × d mm is placed on the XOY plane, with conductivity of $\sigma =3.8\times 10^{7}\;S/m$. The variation of parameter d investigates the response of PEC to plate thickness, and the change of parameter h tests the interference of lift-off variation on the thickness detection.

### 3.1. The D-ATC Curve of the Aluminum Plate

The D-ATC curve is calculated from the induced electromotive force $\varepsilon_{1}\left(t\right)$ of the receiver coil, which is simulated by ANSYS Maxwell3D software, an electromagnetic field simulation tool that finds the distribution of the spatial electromagnetic field and its derivative over time based on the finite element method. The largest side length of each element is limited to 5 mm, and a rectangular current is employed as the excitation with an amplitude of 2 A and edge time of 40 μs. The D-ATC curve of the d=10 mm aluminum plate is shown by the red solid in Figure 2a, and its slope curve is drawn in Figure 2b.

As shown in Figure 2a, after the excitation current is completely turned off (t=0), the D-ATC curves of the aluminum plate present an obvious rising process, and the time required to stabilize the slope of the curve within 0.2 is marked as transient time $T_{tr}$. It can be seen from Figure 2b that the transient time of the aluminum plate is about 4.2 ms. During $t>T_{tr}$, the ascent of the D-ATC curve slows down and enters a relatively steady state, and it is advisable to take $\tau_{ATC}\left(T_{tr}\right)$ as the steady-state value of the D-ATC curve. 

### 3.2. The Relationship between the D-ATC Curve and the Diffusion of ECD

The shape of the D-ATC curve describes the diffusion and attenuation of the eddy current flow in the inspected material. 

The distribution of ECD in the Z0Y section shown in Figure 3a reveals the diffusion of the eddy current in the aluminum plate. Considering the symmetry of the PEC inspection model, only the positive semi-axis part of the Y axis is displayed. It can be seen that the current is not uniformly distributed in the aluminum plate, the relatively concentrated current shown in red forms a current loop, whose decay rate dominates the time constant of the eddy current. 

As shown in Figure 3a, the equivalent current loop diffuses longitudinally to the horizontal axis of the aluminum plate during 0<t≤4 ms, and the cross-section of the current loop gradually expanded. Since the internal resistance *R* of the equivalent current loop in (2) is inversely proportional to its cross-sectional area, the reduced equivalent resistance significantly increases the corresponding D-ATC curve, as shown by the red solid line in Figure 2a. 

It should be noted that the equivalent current loop is constrained on the surface of the aluminum plate during t≤1 ms, as shown in the red area of the t=1 ms subgraph in Figure 3a, the compressed current cross-sectional area restricts the rising speed of the D-ATC curve during this period shown in Figure 2a.

Once the maximum of ECD reaches the horizontal axis of the aluminum plate, as shown in the subgraph of t=4 ms in Figure 3a, the equivalent current loop no longer diffuses along the longitudinal direction. Meanwhile, the ascent of the D-ATC curve in Figure 2 slows down and enters a relatively steady state. 

Therefore, the transient time $T_{tr}$ of the D-ATC curve corresponds to the moment that the maximum of ECD penetrates to d/2 of the aluminum plate, so the thickness parameter d of the aluminum plate can be extracted by the transient time $T_{tr}$.

On the other hand, the radius of the equivalent current loop increases as the maximum of ECD spreads in the horizontal direction, and this radial diffusion phenomenon does not stop with the longitudinal diffusion. As shown in Figure 3a, the radius of the equivalent current loop is expanded from 6 cm (at t=4 ms) to 8.2 cm (at t=14 ms). With the radial diffusion of the equivalent current loop, the amplitude of the D-ATC curve in Figure 2a still rises slowly after $t>T_{tr}$. Therefore, the steady-state interval of the D-ATC curve mainly reflects the radial diffusion process of eddy current.

It should be noted that the ECD is symmetrically distributed about the horizontal axis of the aluminum plate from the moment $t=T_{tr}$, hence the equivalent section radius $r_{c}$ is approximately equal to d/2, which associates the thickness parameter d of the aluminum plate with the steady-state value $\tau_{ATC}\left(T_{tr}\right)$ of the D-ATC curve.

In short, the shape of the D-ATC curve associates the diffusion of ECD with the geometric dimensions of the test pieces. It hints that the thickness of the aluminum plate can be evaluated by the transient time $T_{tr}$ and steady-state value $\tau_{ATC}\left(T_{tr}\right)$ of the D-ATC curve.

## 4. Detection of Plate Thickness

This section investigates the characteristics of aluminum plate thickness on the D-ATC curve.

Set the thickness of the aluminum plate in Figure 1a to d=5 mm, 10 mm, 15 mm, and 20 mm, respectively, and the corresponding D-ATC curves are drawn as the blue dashed line, red solid line, black dotted line, and yellow dash-dotted line in Figure 4. It can be seen that the increase in the thickness prolongs the transient time and increases the steady-state value of the D-ATC curve.

Here we take the ECD profile of the 20 mm thick aluminum plate as an example to further analyze the thickness feature on the D-ATC curve. As shown in Figure 5, the time and color scale of the five subgraphs are consistent with those in Figure 3, but the thickness of the aluminum plate is increased to 20 mm. 

By comparing the two t=1 ms subgraphs, it can be seen that the initial ECD distribution is independent of the thickness. With the same geometric dimensions (e.g., $5.8\times 10^{4}\;A/m^{2}$ contour), the similar equivalent current loops of the two aluminum plates overlap the corresponding D-ATC curves during this period, as shown by the red solid line and the yellow dash-dotted line during t≤1 ms in Figure 4.

As the maximum value of ECD approaches to the depth of d/2, the difference in penetration can be observed in the t=3 ms subgraphs of Figure 3a and Figure 5. It can be seen that the maximum ECD in Figure 3a has penetrated to 4.5 mm, and the ECD is almost symmetrically distributed about the d/2 axis, while the penetration depth of the maximum ECD in Figure 5 is only 3 mm, and the equivalent current loop has not yet separated from the surface of the aluminum plate. With a larger current cross-sectional area, the amplitude of the D-ATC curve shown by the red solid line in Figure 4 is significantly higher than the yellow dash-dotted line during 1 ms<t≤4 ms. This difference in time constant is also reflected in the colors of the two ECD profiles. Although sharing the same color scale, the maximum ECD of the 10 mm aluminum plate is displayed as warm during 1 ms<t≤4 ms, while that of the 20 mm aluminum plate is marked as green, a color that indicates weaker current density, which indicates that the eddy current in the thicker aluminum plate decays faster, so the corresponding D-ATC curve has a smaller amplitude than the thinner aluminum plate during this period.

After t>4 ms, the maximum ECD of the 10 mm aluminum plate no longer penetrates in the longitudinal direction, and the corresponding D-ATC curve enters the steady state. Since the transient time $T_{tr}$ corresponds to the moment when the maximum of ECD penetrates to d/2 of the aluminum plate, the increase in d prolongs the transient time of the D-ATC curves in Figure 4. It can be seen from Figure 5 that the maximum of ECD penetrates to the horizontal axis around t=14 ms. At this time, the cross-sectional area of the current loop is related to the thickness d, so the thicker aluminum plate reduces the loop resistance, thereby increasing the steady-state value of the yellow dash-dotted line in Figure 4. This difference in the steady-state time constant is also reflected in the colors of the two ECD profiles. As shown in the t=20 ms subgraph, the maximum ECD of the 20 mm thick aluminum plate is displayed in warm color, while that of the 10 mm thick aluminum plate is marked in green instead, which further confirms the thicker aluminum plate has a higher $\tau_{ATC}\left(T_{tr}\right)$.

In short, both the transient time and steady-state value of the D-ATC curve can be used to investigate the thickness of the conductive plate, so it is necessary to study the detection sensitivity of the parameters $T_{tr}$ and $\tau_{ATC}\left(T_{tr}\right)$ to the thickness d.

As shown by (4), the penetration depth of the eddy current in a uniform half-space increases with time as power laws, and the exponential coefficient is deduced to 0.5. In this study, the thickness d is fitted with the transient time $T_{tr}$, as shown in Figure 6a, and the calculated exponential coefficient increases to 0.5983 with the sum of squared error (SSE) of $6.194\times 10^{-9}$. This increment is due to the significant difference in the geometric dimensions of the two computation models. In view of the relationship between $T_{tr}$ and d, it can be inferred that the thickness detection of extremely thin non-ferromagnetic conductors will be a challenge for the $T_{tr}$ signature, which will be further tested through experiments.

On the other hand, the thickness d is fitted with the steady-state value $\tau_{ATC}\left(T_{tr}\right)$ of the D-ATC curve, as shown in Figure 6b. It can be seen that $\tau_{ATC}\left(T_{tr}\right)$ increases linearly with d, which makes it suitable as an indicator of the thickness d.

Through the above analysis, it can be seen that both the transient time $T_{tr}$ and steady-state value $\tau_{ATC}\left(T_{tr}\right)$ of the D-ATC curve can be used to investigate the thickness of the conductive plate. Considering the power law between $T_{tr}$ and thickness d, the $T_{tr}$ signature may not have satisfactory detection sensitivity for extremely thin non-ferromagnetic conductors. Meanwhile, the $\tau_{ATC}\left(T_{tr}\right)$ signature looks more suitable as an indicator of thickness due to their linear relationship. 

## 5. The Effect of Lift-Off on the D-ATC Curve

During the inspection process, there may be variation in the distance between the inspection coil probe and the test piece. The lift-off variations can be caused by varying coating thicknesses, irregular sample surfaces or the operator’s movements.

In this section, the signal disturbance caused by large lift-off variation is simulated by adjusting the parameter h shown in Figure 1a. 

Generally, conventional PEC inspection diagnoses the thickness of the test piece based on the difference between the detection signal and the reference signal. Taking the EMF signal of (h, d)=(100, 5) mm model as the reference, the differential EMF signals corresponding to the aluminum plates with d=10 mm and d=15 mm are shown by the red solid line and blue dashed line in Figure 7. When the lift-off drops to h=50 mm, differential EMF signal corresponding to the d=10 mm aluminum plate is redrawn as the green dash-dotted line. It can be seen that the shortening of lift-off not only increases the amplitude of the differential EMF, but also changes the slope and zero crossing time of the signal. Therefore, the varied PEC response caused by the lift-off effect is prone to mask defect signals or generate a false alarm.

The longitudinal diffusion speed of ECD is hardly affected by the lift-off [21]. The D-ATC curves of d=5 mm aluminum plate with h=100 mm and h=50 mm are shown in the black dotted line and green dash-dotted line in Figure 8a, respectively. It can be seen that the shortening of lift-off does not change the time characteristics of the D-ATC curve, and the two curves share the same transient time $T_{tr1}$. Meanwhile, the steady-state value of the D-ATC curve decreases with lift-off, this is because the shortening of the lift-off reduces the radius of the equivalent current loop, and the consequent reduced inductance accelerates the decay speed of the eddy current. 

Still taking the D-ATC curve of (h, d)=(100, 5) mm model as the reference, the D-ATC curves of the 10 mm and 15 mm aluminum plates at the same life-off are compared with it to get the differential D-ATC curve, as shown by the red solid line and blue dashed line in Figure 8b. It can be seen that the transient time $T_{tr2}$ and $T_{tr3}$ of the differential D-ATC curves are consistent with that of the corresponding D-ATC curve shown in Figure 4, and the time to minimum $T_{min}$ of the differential D-ATC curves is slightly ahead of the transient time of the reference D-ATC curve. 

When the lift-off is shortened to h=50 mm, the differential D-ATC curve of the (h, d)=(50, 10) mm model is redrawn as the green dash-dotted line in Figure 8b. It can be seen that the shortening of the lift-off leads to compression of the differential D-ATC curve, but it does not change the transient time $T_{tr2}$ and the time to minimum $T_{min}$. 

In short, the time features of the D-ATC curve are immune to lift-off variations, and the transient time $T_{tr}$ can be used to detect the thickness of aluminum plate under large lift-off conditions.

## 6. Experiment

An experimental study is conducted to further discuss the sensitivity of the D-ATC method to the thickness detection of thin non-ferromagnetic conductors, as well as the adaptability to large lift-off variations.

The experimental setup used in this work is illustrated in Figure 9, the 300 × 300 mm aluminum plates of different thicknesses are placed above the inspection coil probe through insulating partitions. The inspection coil probe is formed by a central loop device, the outer coil, of 30 mm outside diameter, was used for excitation, while the inner one, of 20 mm outside diameter, was used for sensing. The lift-off can be switched between h=50 mm or h=100 mm by adjusting the number of insulating partitions. A bipolar pulse current with amplitude of 3 A and edge time of 20 μs was employed as the pulsed excitation, and the sampling frequency was set to 1.25 MHz. 

In the case of lift-off h=100 mm, the D-ATC curves of aluminum plates with thicknesses d=1 mm, 3 mm, and 5 mm are shown as the blue dashed line, red dotted line, and yellow dash-dotted line in Figure 10, respectively. It can be seen that the steady-state values $\tau_{ATC}\left(T_{tr}\right)$ of the three D-ATC curves obey the arithmetic progression, which verifies the linear relationship between the thickness d and $\tau_{ATC}\left(T_{tr}\right)$ shown in Figure 6b. 

On the other hand, the transient time $T_{tr1}$ and $T_{tr2}$ corresponding to the 1 mm and 3 mm thick aluminum plates are clearly distinguishable, which is deduced to be a potential challenge, because the power law shown in Figure 6a reduces the resolution of $T_{tr}$ for the thin plates. These characteristics are consistent with the simulation results in Section 4. Therefore, the $T_{tr}$ and $\tau_{ATC}\left(T_{tr}\right)$ signatures have sub-millimeter resolution for thickness variation.

Note that the conductivity of the tested 1060 aluminum plate is similar to that in simulation, but the D-ATC curve corresponding to d=5 mm in Figure 10 presents a longer transient time than that of Figure 4, this may be caused by the limitations of finite element division. Since the thickness of the aluminum plate is much smaller than its side length, the maximum side length of the finite element mesh is set to 5 mm by weighing the calculation resources and accuracy. Therefore, the investigation features of the proposed D-ATC method for thicknesses above 5 mm are displayed by simulation, and the applicability for thicknesses below 5 mm is verified by experiments. Nevertheless, their response characteristics to the thickness variations are consistent.

In order to test the stability of $T_{tr}$ signature in large lift-off variations, the lift-off was reduced to h=50 mm by removing one insulating partition, and PEC inspections were repeated on the above three aluminum plates to observe the lift-off effect on the proposed D-ATC method. Taking the D-ATC curve of the 3 mm aluminum plate as the reference, the differential D-ATC curve of d=5 mm aluminum plate is drawn as the blue dotted line in Figure 11, and the corresponding result with h=100 mm is drawn as the red dashed line.

It can be seen from Figure 11 that the shortening of lift-off improves the signal-to-noise ratio of the D-ATC curve and compresses the amplitude of the differential D-ATC curve. However, lift-off variations did not change the time features (e.g.,$\;T_{min}$ and $T_{tr3}$) of the differential D-ATC curve, which is consistent with the simulation result in Figure 4b. Therefore, the $T_{tr}$ signature of the D-ATC curve is not affected by lift-off variations.

This section further studies the characteristics of D-ATC on conductor thickness. The experimental results show that the $T_{tr}$ and $\tau_{ATC}\left(T_{tr}\right)$ signatures have sub-millimeter resolution for thickness detection, and the $T_{tr}$ signature is immune to lift-off variations.

## 7. Conclusions

The pulsed eddy current in the test piece diffuses similar to a smoke ring while decays under the action of the conductor resistance, by drawing lessons from the zero-input response of the circuit theory, this paper extracts the dynamic time constant of the equivalent current loop from the PEC response, the proposed D-ATC method associates the dynamic behavior of the induced eddy current with the geometric dimensions of the test pieces, thus the thickness of aluminum plate can be evaluated. As a normalized decay rate indicator, the criterion based on the time characteristic of the D-ATC curve is immune to lift-off variations, and it can be used to detect the sub-millimeter thickness of non-ferromagnetic conductive plates under large lift-off conditions.

## Figures and Tables

**Figure 1 sensors-22-00614-f001:**
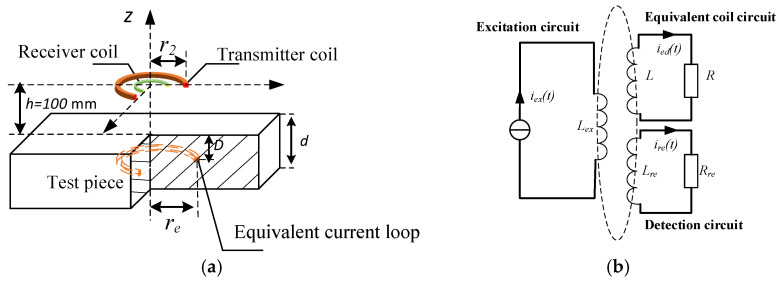
The coil-coupling model (CCM) of the PEC inspection. (**a**) The CCM applies an equivalent current loop with finite dimensions to approximate the general eddy current effect of the specimen. (**b**) The air-core transformer model, in which the primary side is the excitation circuit, and the secondary side is the equivalent current loop modeled by an inductor *L* and a resistor *R* in series connection.

**Figure 2 sensors-22-00614-f002:**
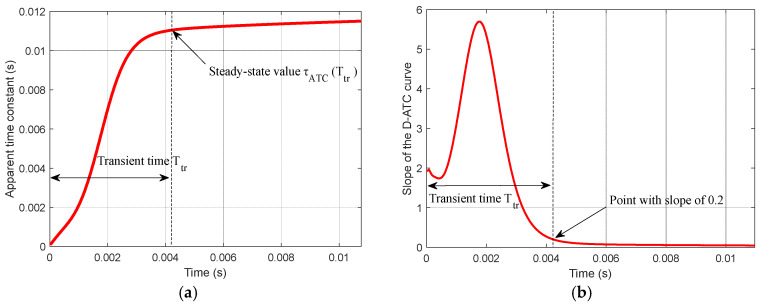
The D-ATC curve and its slope curve of the aluminum plate, the sampling step length is 20 μs. (**a**) The D-ATC curve presents an obvious rising process, after the transient time $T_{tr}$, after that the ascent of the D-ATC curve slows down and enters a relatively steady state. (**b**) The time required to stabilize the slope of the D-ATC curve within 0.2 is marked as transient time $T_{tr}$, and the amplitude of the curve at this time is recorded as the steady-state value.

**Figure 3 sensors-22-00614-f003:**
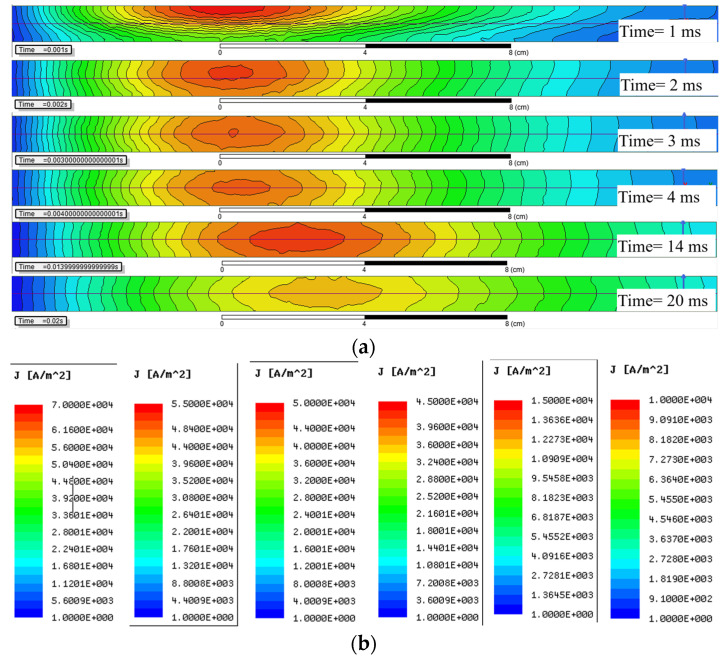
The distribution of eddy current density (ECD) in an aluminum plate with *d* = 10 mm. (**a**) ECD in the Z0Y section diffuses similar to a smoke ring during the interval of pulse excitation. (**b**) The color scale corresponding to the time shown in (**a**), the minimum value of ECD is limited to 1 $A/m^{2}$, and the maximum values are set to $7\times 10^{4}A/m^{2}$, $5.5\times 10^{4}A/m^{2}$, $5\times 10^{4}A/m^{2}$, $4.5\times 10^{4}A/m^{2}$, $1.5\times 10^{4}A/m^{2}$ and $1\times 10^{4}A/m^{2}$, respectively.

**Figure 4 sensors-22-00614-f004:**
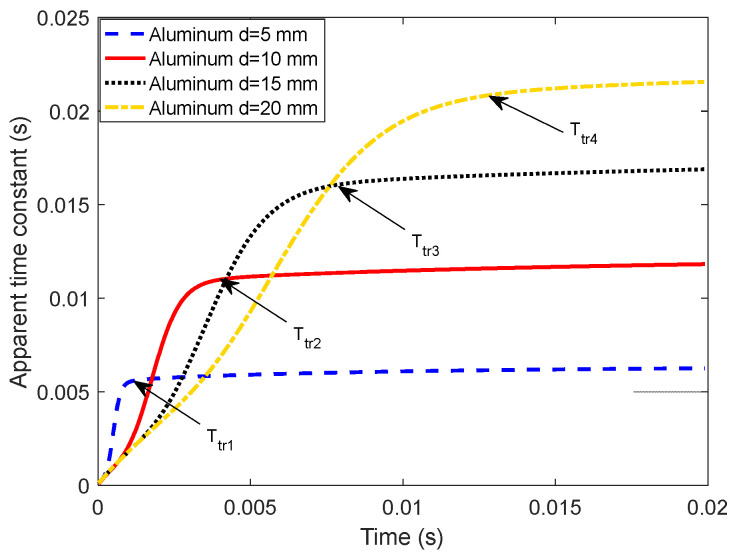
Characterization of D-ATC curve to the thickness of aluminum plate with a large lift-off of 100 mm. The increase in the thickness of the aluminum plate prolongs the transient time of the D-ATC curves and improves its steady-state value.

**Figure 5 sensors-22-00614-f005:**
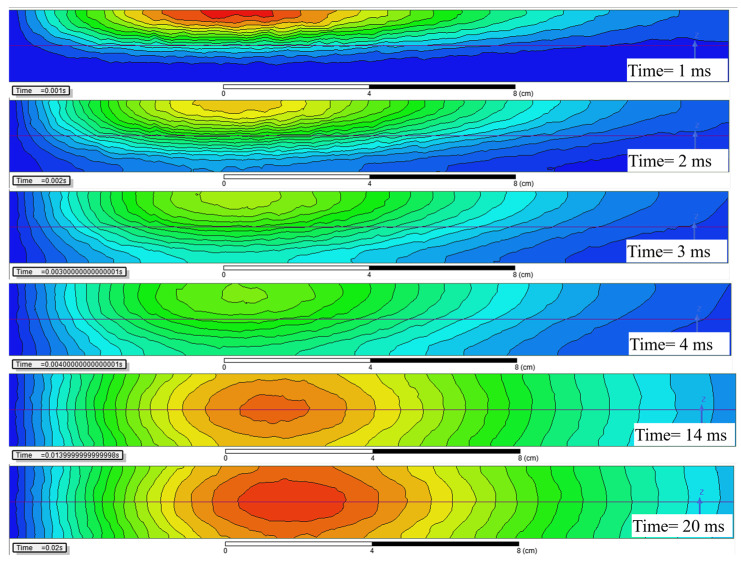
The distribution of eddy current density (ECD) in a 20 mm thick aluminum plate with the same color scales shown in Figure 3b, in which the maximum values of ECD are set to 70 $\mathrm{kA}/m^{2}$, 55 $\mathrm{kA}/m^{2}$, 50 $\mathrm{kA}/m^{2}$, 45 $\mathrm{kA}/m^{2}$, 15 $\mathrm{kA}/m^{2}$ and 10 $\mathrm{kA}/m^{2}$, respectively.

**Figure 7 sensors-22-00614-f007:**
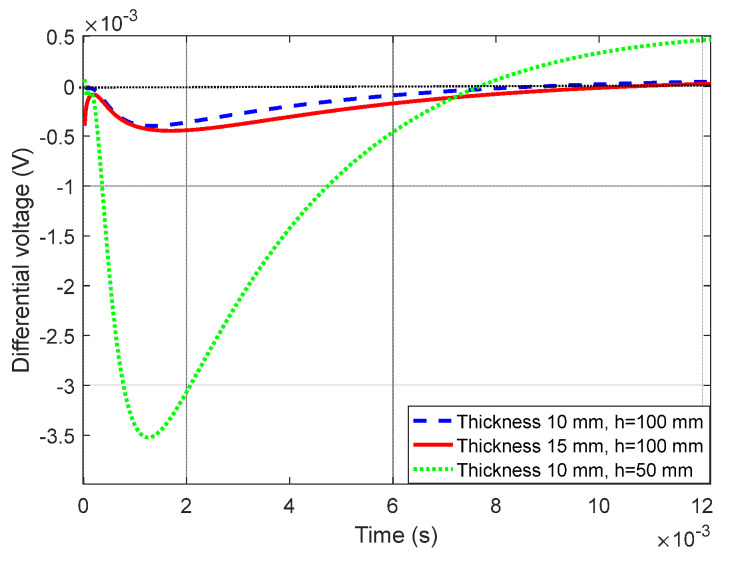
Differential induced voltage curve. The shortening of lift-off not only increases the amplitude of the differential EMF, but also changes the zero crossing time of the signal.

**Figure 8 sensors-22-00614-f008:**
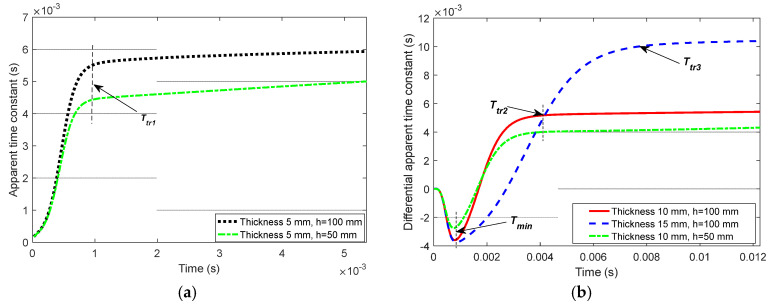
The effect of lift-off distance on pulsed eddy current detection signal. (**a**) The D-ATC curves corresponding to d=5 mm aluminum plate with the lift-off of h=100 mm and h=50 mm, which have similar shapes and almost the same transient time. (**b**) Differential D-ATC curves of the 10 mm and 15 mm aluminum plates with the lift-off of h=100 mm and  h=50 mm, lift-off variations did not change the transient time $T_{tr2}$ and the time to minimum $T_{min}$.

**Figure 9 sensors-22-00614-f009:**
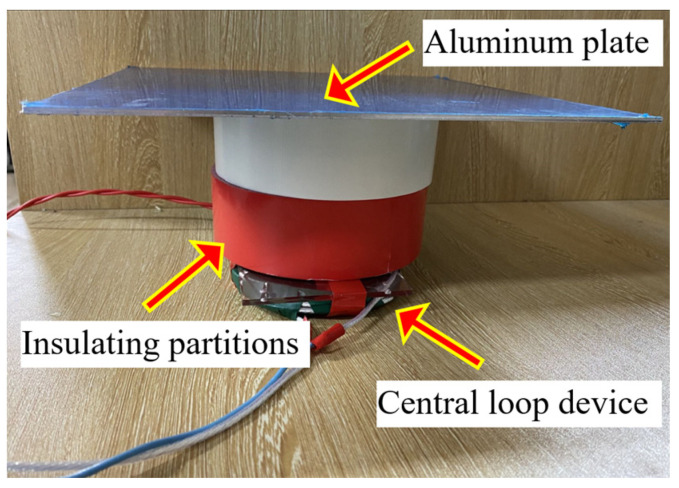
PEC test system for aluminum plate thickness detection. Aluminum plates of different thicknesses are placed above the inspection coil probe through non-magnetic insulating partitions, and the lift-off can be set to *h* = 50 mm or *h* = 100 mm by adjusting the number of insulating partitions.

**Figure 6 sensors-22-00614-f006:**
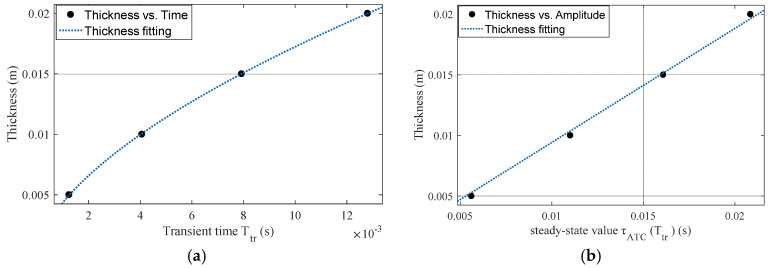
Fitting results of aluminum plate thickness. (**a**) The relationship between the thickness of the aluminum plate and the transient time follows the power law. (**b**) The thickness of the aluminum plate has a linear relationship with the steady-state value of the D-ATC curve.

**Figure 10 sensors-22-00614-f010:**
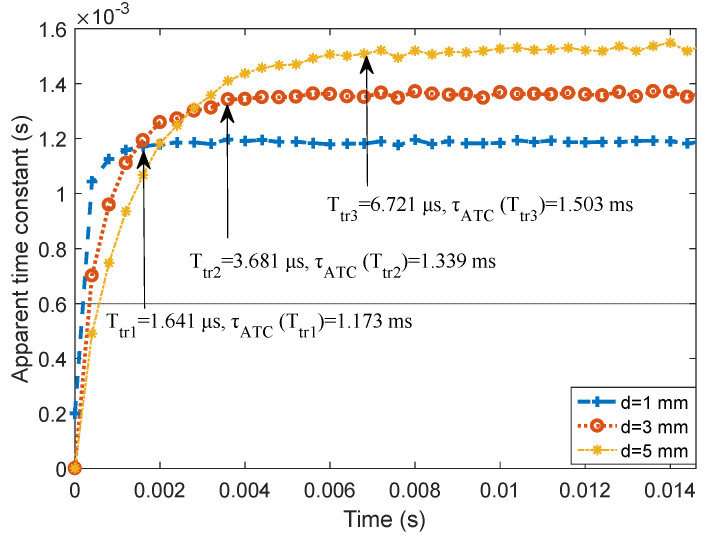
Measured D-ATC curve of three thickness aluminum plates with 100 mm lift-off. There is a cross between two D-ATC curves with different thicknesses, the steady-state value of the D-ATC curve is positively correlated with the thickness of the aluminum plate, and the transient time of the D-ATC curve extends with the increase of specimen thickness.

**Figure 11 sensors-22-00614-f011:**
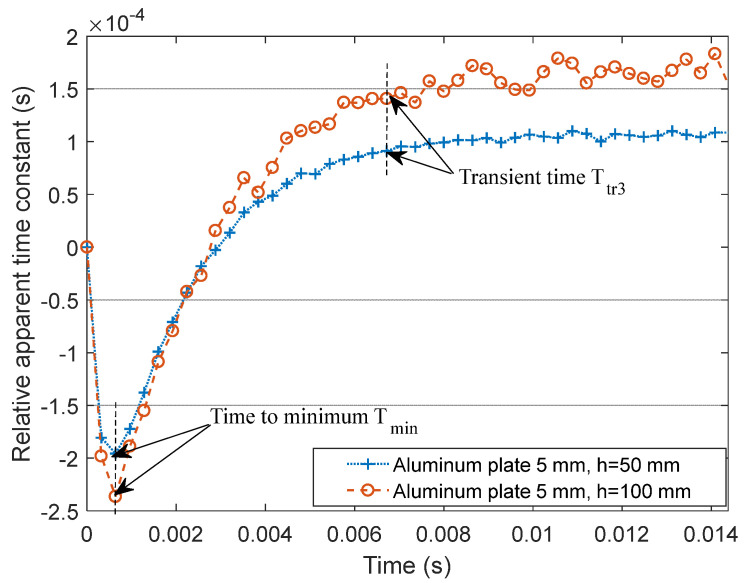
The effect of lift-off variations on the differential D-ATC curve. The increase in lift-off reduces the signal-to-noise ratio of the D-ATC curve but does not change the time characteristics.

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
