# Peer review of "A Novel Pulsed Eddy Current Criterion for Non-Ferromagnetic Metal Thickness Quantifications under Large Liftoff"

_sensors, 2022, doi:10.3390/s22020614_

Round 1

Reviewer 1 Report

The draft shows analytical and experimental study on identifying the thickness of the non-metalic sheet structure using PEC. The conventional PEC has been applied to metallic materials for defect detection and thickness evaluations. The presented results maybe of possible interests for readers who are interested in the application of PEC to non-metallic material. However, in the current configuration, this reviewer was not able to find original contribution. For publication as a separate article, it is required to show limitations of the conventional methods when applied to the non-metalic structures. In the current presentation, the presented results look very similar to the conventional pulsed eddy current methology. The claimed originality lies on the proposition of the apparent time constant, but only shows limited advancement compared to the previous approaches listed in the literature review. Other minor comments are

  1. Equation (1) includes error (possibly typing error?)
  2. Equation (3) shows the APC. It is required to show it more specifically, for example, range of integration or how the integration was performed numerically.
  3. Figure 7 shows the current loop device. Since the numerical simulation was performed, the schematic of the current loop device needs to be detailed.
  4. Some physical interpretation of the ATC is required to be included.

Reviewer 2 Report

Review of the manuscript "A novel pulsed eddy current criterion for nonmagnetic metal 2 thickness quantifications" by Haowen Wang, Jiangbo Huang, Longhuan Liu, Shanqiang Qin, Zhihong Fu.

The paper proposes a new method for quantifying metal thickness based on the dynamic apparent time constant (D-ATC) of the full-stage of the PEC response

The work is certainly interesting, but there are a number of comments:

  1. In the introduction, it is necessary to write the relevance of the research, how the method proposed by the authors can be used.
  2. The volume of the article is small, so it makes no sense to give a brief description of the sections in lines 57-64.
  3. If I understand correctly, line 78 is not a system of equations, but a transformed expression. Therefore, they need to be numbered separately (1) and (2).
  4. There are explanations not for all terms of the equations. What is M, t?
  5. In the description of the device (section 3), it is necessary to add the technical characteristics of the equipment used.
  6. In fig. 3 it is necessary to add explanatory labels. Those that exist are very small.
  7. Section conclusion - definitely needs to be supplemented, since in this form, it does not reflect all the results obtained in the work.
  8. Why the modeling in Section 4 for plate thicknesses of 5, 10 and 15 mm, and in the experimental section - for 1, 3 and 5 mm thicknesses? And even for 5 mm the convergence of the experimental and the model "dynamic apparent time constant" is low.
  9. The sensitivity of the method proposed by the authors is not indicated. If it is intended to be used for the determination of corrosion (as indicated in section 4), then the method should provide a fixation of dimensional changes up to 1 mm. This also needs to be analyzed.
  10. Is it possible to use this method for magnetic materials?

In general, I think that the work needs to be improved.

Round 2

Reviewer 1 Report

The reviewer comments have been rebutted with suitable revisions. 

Reviewer 2 Report

Many thanks to the Authors for revising the manuscript. This actually turned out to be another article, but in this form it is quite worthy of publication in the journal.